# Detection of *Culex tritaeniorhynchus* Giles and Novel Recombinant Strain of Lumpy Skin Disease Virus Causes High Mortality in Yaks

**DOI:** 10.3390/v15040880

**Published:** 2023-03-29

**Authors:** Yan Li, Zhibo Zeng, Kewei Li, Mujeeb Ur Rehman, Shah Nawaz, Muhammad Fakhar-e-Alam Kulyar, Miao Hu, Wenqian Zhang, Zhao Zhang, Miao An, Jianwu Hu, Jiakui Li

**Affiliations:** 1College of Veterinary Medicine, Huazhong Agricultural University, Wuhan 430070, China; yanli.vet@webmail.hzau.edu.com (Y.L.); fakharealam786@hotmail.com (M.F.-e.-A.K.);; 2Disease Investigation Laboratory, Livestock & Dairy Development Department, Zhob 85200, Pakistan; 3Agriculture and Rural Affairs Department, Tibet Autonomous Region Veterinary Bureau, Lhasa 850000, China

**Keywords:** yak, lumpy skin disease virus, vaccine-related recombinant, *Culex tritaeniorhynchus* Giles

## Abstract

Lumpy skin disease virus (LSDV) is capable of causing transboundary diseases characterized by fever, nodules on the skin, mucous membranes, and inner organs. The disease may cause emaciation with the enlargement of lymph nodes and sometimes death. It has had endemic importance in various parts of Asia in recent years, causing substantial economic losses to the cattle industry. The current study reported a suspected LSDV infection (based on signs and symptoms) from a mixed farm of yak and cattle in Sichuan Province, China. The clinical samples were found positive for LSDV using qPCR and ELISA, while LSDV DNA was detected in *Culex tritaeniorhynchus* Giles. The complete genome sequence of China/LSDV/SiC/2021 was determined by Next-generation sequencing. It was found that China/LSDV/SiC/2021 is highly homologous to the novel vaccine-related recombinant LSDV currently emerging in China and countries surrounding China. Phylogenetic tree analysis revealed that the novel vaccine-associated recombinant LSDV formed a unique dendrograms topology between field and vaccine-associated strains. China/LSDV/SiC/2021 was found to be a novel recombinant strain, with at least 18 recombination events via field viruses identified in the genome sequence. These results suggest that recombinant LSDV can cause high mortality in yaks, and its transmission might be due to the *Culex tritaeniorhynchus* Giles, which acts as a mechanical vector.

## 1. Introduction

Lumpy skin disease virus (LSDV) is primarily transmitted through flies, mosquitoes, and ticks of the phylum Arthropoda [1]. It is a critical transboundary disease that causes clinical symptoms characterized by pox eruption on the skin, mucous membranes, and internal organs of cattle, giraffes, Asian water buffalo (Bubalus bubalis), impalas, and many other animals [2,3]. Animals infected with LSDV pose a substantial economic threat to the world’s cattle industry and other ruminants [4].

In 1929, when LSDV was first demonstrated in Zambia, its occurrence was limited to being popular in Africa [5]. Since the beginning of the 20th century, LSDV has expanded from Africa to the Middle East, Europe, and Asia [6]. In 2015–2018, for protection against LSDV, Balkan countries, and Kazakhstan chose to use a homologous live attenuated vaccine based on the Neethling strain, while Armenia and Russia chose the sheep pox virus heterologous vaccine for immunization [6,7,8,9]. During this period, cases of natural infection caused by the vaccine-associated recombinant strain LSDV/Russia/Saratov/2017, the Kenyan KSGPO-240/Kenya/1958, and Neethling/LW-1958 live attenuated vaccine (LAV) strains were reported in Russia on the border with Kazakhstan [10]. This recombination may have occurred due to the illegal use of the homologous vaccine in Russia or the unlawful transfer of Lumpivax-vaccinated (KEVEVAPI) animals from Kazakhstan [11]. Since then, recombinant strains of LSDV have continued to expand in Asia.

A novel LSDV strain emerged in China’s Xinjiang province bordering Kazakhstan in 2019 [12]. Subsequently, comparable novel recombinant strains of LSDV were reported from Vietnam, Thailand, and Mongolia (Figure 1) [13]. The emerging LSDV recombinant strains in Asia have a distinctly different topology from the LSDV/Russia/Saratov/2017 strain in the phylogenetic trees [5,7,13].

In October 2021, an outbreak of suspected LSDV occurred on a mixed yak and cattle farm in Sichuan province, China. The outbreak resulted in high morbidity and mortality rates in yaks. This study is the first to describe that a novel strain of LSDV can cause yak morbidity and mortality because of *Culex tritaeniorhynchus* Giles, which may be an essential source of LSDV transmission. The study provides a vital information reference for LSDV as a serious transboundary pathogen.

## 2. Materials and Methods

### 2.1. Sampling

Clinical samples were collected from yaks suffering high fever, skin nodules, enlarged lymph nodules, oculonasal discharge, and respiratory distress at a mixed yak and cattle farm (21 cattle and 15 yaks) in Sichuan, China, in October 2021. Clinical samples included fecal swabs (n = 4, 4 yaks), nasal swabs (n = 4, 4 yaks), whole blood (n = 4, 4 yaks), serum (n = 28, 21 cattle and 7 yaks), skin nodules of diseased yaks (n = 3), and the *Culex tritaeniorhynchus* Giles (n = 16). After blood collection, the serum was kept at room temperature for 1 h to extract the serum. Then, the centrifugation was performed at 6000 rpm for 5 min to remove coagulated red blood cells before storing them in a refrigerator at −20 °C. Other samples were immediately immersed in a phosphate buffer solution for DNA extraction.

### 2.2. Detection of LSDV DNA

According to the manufacturer’s instructions, viral genomic DNA was extracted from collected samples using the Virus DNA/RNA Extraction Kit 2.0 (Vazyme, Nanking China). Development of a SYBR Green I quantitative real-time PCR (qPCR) assay with specific primers for detection of viral GPCR genes (forward primer: 5′-AGTCGAATATAAAGTAATCAGTC-3′, reverse primer: 5′-CCGCATA-TAATACAACTTATTATAG-3′) [14]. Briefly, the amplification reaction was performed in a 25 μL final volume with the following components: 2 μL DNA; 0.75 μL of each 10 μM primer; 12.5 μL 1 × T.B. green Premix Dimer Eraser (Takara, Dalian, China); and an appropriate volume of ddH2O up to 25 μL. The qPCR profile condition was as follows: 95 °C for the 30 s; 40 cycles of 95 °C for 5 s, 55 °C for 30 s, and 72 °C for 30 s.

### 2.3. Laboratory Test of LSDV Antibodies

Anti-LSDV antibodies in the serum samples were detected using the anti-LSDV antibody ELISA kit (LSBIO, Lanzhou, China) according to the manufacturer’s instructions. Serum samples with a P.I. ratio ≥ 55% were considered to be positive for antibodies to the LSDV antigen.

### 2.4. Whole Genome Sequencing

Whole genome sequencing was performed on the Illumina Hiseq2000 (Illumina, CA, USA) platform. To assemble the genome, reads were mapped to the reference genome (LSDV/GD01/China/2020, MW355944) [15]. The assembled high-quality whole genome sequence (China/LSDV/SiC/2021) was stored in the GenBank database with the accession number OP654649.

### 2.5. Phylogenetic Analysis

Sequence alignment was performed by the MAFFT v7.505 [16]. Maximum-likelihood dendrograms from LSDV complete sequences (using the K3Pu+F+R2 model) were determined by IQ-TREE in PhyloSuite v1.2.2 [17]. The values on the branches are the percentage of 1000 bootstraps supporting the branching pattern.

### 2.6. Restructuring Event Detection

For the analysis of recombination events based on LSDV genomic sequences, seven methods from the Recombination Detection Program (RDP v4.101) software package (RDP, GENECONV, Chimera, MaxChi, BootScan, SiScan, and 3Seq) were used for recombination analysis [5]. The events were further characterized by Simplot v3.5.1 [18].

## 3. Results and Discussion

At the time of sample collection, yaks had a fever, skin lesions, and swollen superficial lymph nodes. The disease lasted 3–5 days, with a mortality rate of 53.33% (8/15) and 22.22% (8/36) in yaks and cattle, respectively. The recovered yaks and cattle had crusts on their skin. Clinical findings showed skin and mucous membranes were covered with pox lesions, forming scabs (Figure 2A–D). Pox lesions mainly appeared on the mammary glands, less hairy neck parts, edges of the nasal cavity, hairless upper eyelids, and scrotum (Figure 2A–E). Later, severe pox lesions scab fell off to form open ulcers (Figure 2F).

Among the different samples, the highest concentration of LSDV DNA was found in skin nodules [14]. In this study, the virus concentration was higher in fecal swabs and skin nodules, with a positive rate of 75% and 100%, respectively (Figure 3). In previous reports, LSDV caused 0.96%–14.56% mortality in cattle [19,20]. Serological results indicated that 28 serums (21 cattle and 7 yaks) have P.I. > 55%. The farms in the area were not immunized with the Lumpivax vaccine, but the presence of LSDV antibodies indicated previous exposure to the field strain of LSDV. All cattle and yaks on the farm were infected with LSDV, confirming a 100% LSDV infection rate (28/28).

*Culex tritaeniorhynchus* Gile, a member of the mosquito family mainly distributed in Asia, acts as a significant transboundary carrier of infectious pathogens [21]. Laboratory analysis has revealed that *Aedes aegypti* and *Culex quinquefasciatus* can carry and mechanically transmit LSDV [1]. The current study found that *Culex tritaeniorhynchus* Giles had a higher copy count of LSDV DNA than whole blood (Figure 3), indicating its potential role in transmitting LSDV. This may be related to the widespread distribution of *Culex tritaeniorhynchus* Giles in Asia, carrying LSDV.

According to phylogenetic trees (Figure 4), LSDV occurring in China and reference strains has a distinct topological clustering of field vaccine and recombinant-associated strains. The LSDV strain of yak origin was a branch of recombinant-associated strains between field and vaccine-associated strains, forming a new branch with the currently reported clustering of LSDV strains from China [7,14]. Notably, the LSDV strains prevalent in China and neighboring countries have the same dendrograms topology, identified as novel recombinant strains in the current study.

Recombination events with statistically significant support (*p* ≤ 0.01) were counted for at least three methods in RDP4, as shown in Appendix A. The outcome showed 18 recombination events, with the Kenya strain (MN072619) as the major parent contributing to the recombination of the China/LSDV/SiC/2021 strain. Vaccine-associated strains L.W. 1959, SPPV, and GTPV, were also involved in recombining China/LSDV/SiC/2021. Further calculations of the Kenya strain with the LSDV vaccine-related strain L.W./1959 using Simplot v3.5.1 demonstrated that the China/LSDV/SiC/2021 strain was recombinant (Figure 5).

In previous reports of LSDV infection at cattle and yaks farm in China, morbidity and mortality outbreaks ranged from 6.6% to 100% and 0% to 16.7%, respectively [12,14]. According to our field observations, laboratory diagnostics, phylogenetic analysis, and recombination events, the vaccine-associated high recombinant LSDV infection was noticed in yaks. Such high morbidity and mortality rates demonstrate yaks as the more susceptible species to LSDV infection. No cases of LSDV infection in yaks have been reported since this report, and the morbidity and mortality rate of LSDV infection in yaks was also unknown [2]. Importantly, the current study found that yaks infected with LSDV vaccine-associated strains exhibited high mortality rates with the tendency to spread through *Culex tritaeniorhynchus* Giles. Hence, the spread of LSDV in the country must be addressed for timely control. Moreover, further research is required to demonstrate the susceptibility and incidence of the novel LSDV in yaks.

## Figures and Tables

**Figure 1 viruses-15-00880-f001:**
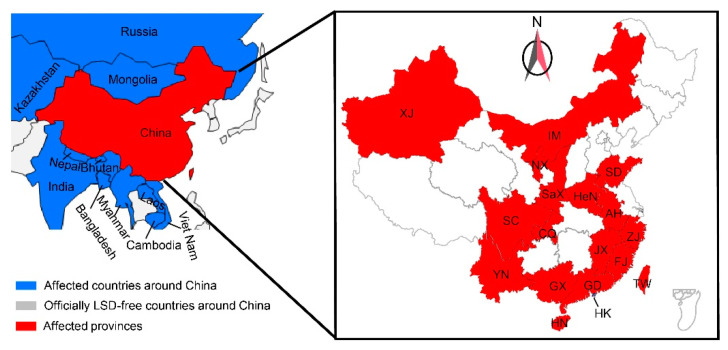
The impact of LSDV in China and its surrounding countries from 2015 to 2022 (Source: https://wahis.oie.int, http://www.customs.gov.cn/, and http://www.moa.gov.cn/gk/sygb/, accessed on 1 September 2022). The occurrence of LSDV in China’s neighboring countries is marked in blue and grey. While red represents provinces in China where LSDV cases were recorded.

**Figure 2 viruses-15-00880-f002:**
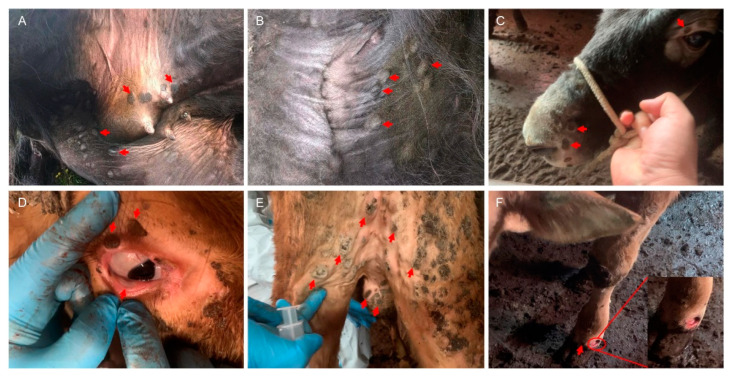
Clinical signs of LSDV infection in yaks and cattle. The red arrow indicates skin nodules or open sores. (**A**–**C**) refers to yak. (**D**–**F**) refers to cattle.

**Figure 3 viruses-15-00880-f003:**
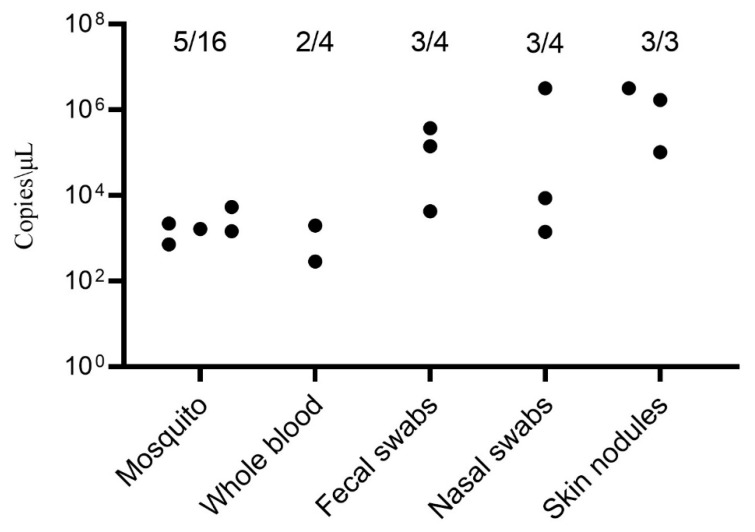
Quantification of LSDV DNA from collected clinical samples using quantitative real-time PCR (qPCR). The numerical value above each group denotes the number of positive results out of the total number of conducted assays. Each group’s positive samples are depicted as black dots that are representing their respective copies/µL.

**Figure 4 viruses-15-00880-f004:**
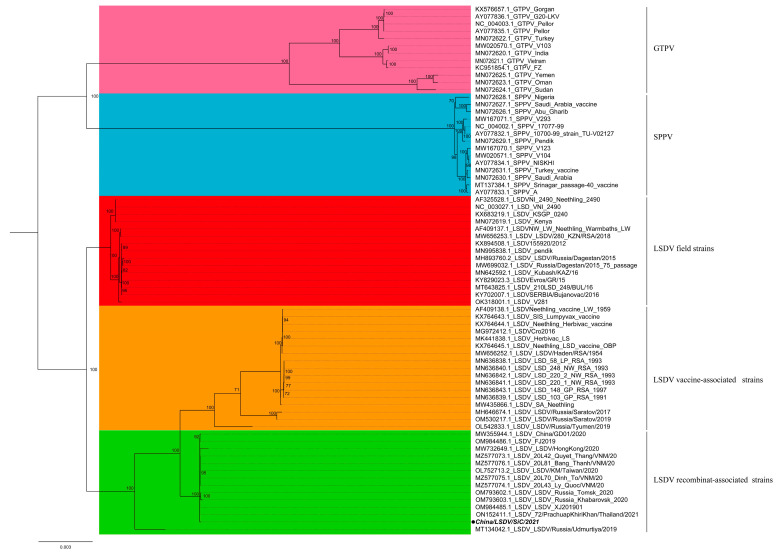
The IQ-TREE v 1.6.8 in the PhyloSuite package was used to establish the evolutionary relationship of SiC/2021 to representative sequences of the reference strain. Visualization output using figtree v1.4.4 (http://tree.bio.ed.ac.uk/software/Figtree/, accessed on 12 October 2022). LSDV isolates are indicated by black dots, while different colors represent different clades.

**Figure 5 viruses-15-00880-f005:**
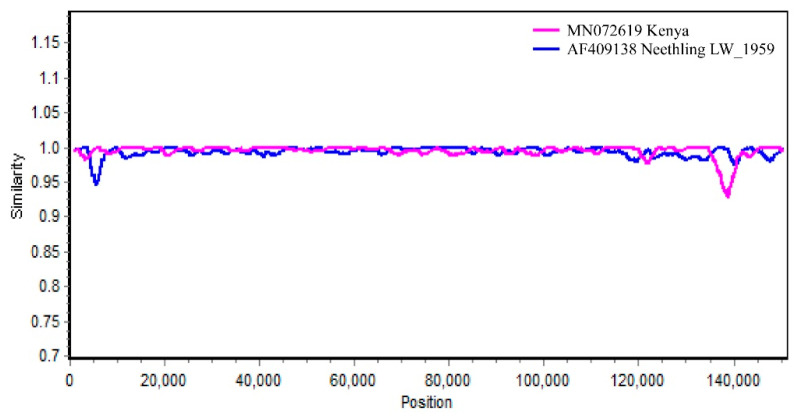
SimPlot v3.5.1 analysis based on the nucleotide sequence of the LSDV genome. Kenya and LW_1959 strains were tested as reference strains.

## Data Availability

The data that support the findings of this study are available on request from the corresponding author.

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
