# Peer review of "Detection of Culex tritaeniorhynchus Giles and Novel Recombinant Strain of Lumpy Skin Disease Virus Causes High Mortality in Yaks"

_viruses, 2023, doi:10.3390/v15040880_

Round 1

Reviewer 1 Report (Previous Reviewer 3)

Review: Detection of Culex Tritaeniorhynchus Giles and Novel Recombinant Strain of Lumpy Skin Disease Virus Causing High Mortality in Yaks

The authors have addressed the issues raised adequately  and therefore the article is suited for publication. However I have still some minor remarks.

Line 42-46: As the sentence is written now, it is understood that both the recombinant and KSGPO and the Neethling vaccine caused natural infection. please rephrase.

Line 67-75: for some samples it’s not clear that they were originated from cattle or yaks for example for the nasal swabs

Author Response

The authors have addressed the issues raised adequately and therefore the article is suited for publication. However I have still some minor remarks.

  • Line 42-46: As the sentence is written now, it is understood that both the recombinant and KSGPO and the Neethling vaccine caused natural infection. please rephrase.

Response: Respected reviewer, we have rephrased the sentence for better understanding as follows:

During this period, cases of infection caused by the vaccine-associated recombinant strain LSDV/Russia/Saratov/2017, the Kenyan KSGPO-240/Kenya/1958, and Neethling/LW-1958 live attenuated vaccine (LAV) strains were reported in Russia on the border with Kazakhstan

  • Line 67-75: for some samples it’s not clear that they were originated from cattle or yaks for example for the nasal swabs

Response: Respected reviewer, We have revised the sentence regarding the collection of samples. The following lines have been added to the revised manuscript.

Clinical samples included fecal swabs (n=4, 4 yaks), nasal swabs (n=4, 4 yaks), whole blood (n=4, 4 yaks), serum (n=28, 21 cattle and 7 yaks), skin nodules of diseased yaks (n=3), and the Culex tritaeniorhynchus Giles (n=16).

Finally, we appreciate your contribution to reviewing our manuscript.

Reviewer 2 Report (Previous Reviewer 1)

Second review: The quality of the manuscript is much better now, although thorough editing of English language is still needed. I would recommend that the authors will let a native English-speaking person to work on it. There are also still other concerns that need to be addressed.

Other comments:

Line 13: First sentence: LSDV can cause only one disease, not several. Please correct.

Lines 24-25:  Suggestion to edit the last sentence of the abstract: “These results suggest that novel vaccine-associated recombinant LSDV can cause high mortality in yaks and Culex tritaeniorhynchus Giles mosquitos may mechanically transmit the virus.

Line 32-34: It is not known if LSD causes all these clinical signs wild ruminants. It would be safer to indicate that these other species can be infected. Also, do not use capital letters cattle, giraffe, add here Asian water buffalo (Bubalus bubalis), impalas…

Line 39: please, check the years 2017-18, I think the Balkans started to vaccinate earlier.

Lines 124-125: “The farm and cattle farms…. “ sentence needs rewriting

Lines 133: “as” should probably been “is”

Lines 136-139: The meaning of the sentence is not clear, it needs to be rewritten.

Lines171-173: Unfortunately, I do not agree with the last part of this sentence “Importantly, this study found that yaks infected with LSDV vaccine-associated strains exhibited high mortality rates with the tendency to spread through Culex tritaeniorhynchus Giles” I would suggest that the authors indicate only that these mosquitos may play a role in the transmission the LSD virus.

Author Response

The quality of the manuscript is much better now, although thorough editing of English language is still needed. I would recommend that the authors will let a native English-speaking person to work on it. There are also still other concerns that need to be addressed.

Other comments:

  • Line 13: First sentence: LSDV can cause only one disease, not several. Please correct.

Response: Respected reviewer, we have modified the sentence as follows:

Lumpy skin disease virus (LSDV) is capable of causing transboundary diseases, characterized by fever, nodules on the skin, mucous membranes and inner organs. The disease may cause emaciation with the enlargement of lymph nodes, and sometimes death.

  • Lines 24-25: Suggestion to edit the last sentence of the abstract: “These results suggest that novel vaccine-associated recombinant LSDV can cause high mortality in yaks and Culex tritaeniorhynchus Giles mosquitos may mechanically transmit the virus.

Response: Thank you for your concern, we have revised the sentence as follows:

These results suggest that recombinant LSDV can cause high mortality in yaks, and its transmission might be due to the Culex tritaeniorhynchus Giles, which acts as a mechanical vector.

  • Line 32-34: It is not known if LSD causes all these clinical signs in wild ruminants. It would be safer to indicate that these other species can be infected. Also, do not use capital letters cattle, giraffe, add here Asian water buffalo (Bubalus bubalis), impalas…

Response: Respected reviewer, as described in Reference no. 2, the wild ruminants such as Thomson's gazelle, giraffe (Giraffe camelopardalis), oryx (Oryx gazelles), springbok (Antidorcas marsupialis), Arabian oryx (Oryx leucoryx), and impala (Aepyceros melampus) can be infected with LSDV. A case of natural infection of a giraffe (Giraffa camelopardalis) in Thailand with clinical signs of pimple rashes on the skin was described according to reference 3. These reports indicate that LSDV poses a significant threat to wild ruminants. Moreover, keeping your point, we have revised the sentence according to your suggestion.

  • Line 39: please, check the years 2017-18, I think the Balkans started to vaccinate earlier.

Response: Respected reviewer, we appreciate your concern and apologize for the mistake. The period was 2015–2016; we have modified the text with the following reference.

Gubbins S, Stegeman A, Klement E, Pite L, Broglia A, Cortiñas Abrahantes J. Inferences about the transmission of lumpy skin disease virus between herds from outbreaks in Albania in 2016. Prev Vet Med. 2020 Aug;181:104602. doi: 10.1016/j.prevetmed.2018.12.008

  • Lines 124-125: “The farm and cattle farms…. “ sentence needs rewriting

Response: Respected reviewer, we have revised the sentence as follows:

The farms in the area were not immunized with Lumpivax vaccine, but the presence of LSDV antibodies indicated previous exposure to the field strain of LSDV.

  • Lines 133: “as” should probably been “is”

Response: Suggestion has been incorporated.

  • Lines 136-139: The meaning of the sentence is not clear, it needs to be rewritten.

Response: Respected reviewer, we have rewritten the sentence as follows.

In the present study, it was found that Culex tritaeniorhynchus Giles had a higher copy count of LSDV DNA than whole blood (Figure 3), indicating its potential role in the transmission of LSDV.

  • Lines171-173: Unfortunately, I do not agree with the last part of this sentence “Importantly, this study found that yaks infected with LSDV vaccine-associated strains exhibited high mortality rates with the tendency to spread through Culex tritaeniorhynchus Giles” I would suggest that the authors indicate only that these mosquitos may play a role in the transmission the LSD virus.

Response: Respected reviewer, Sanz-Bernardo demonstrated that mosquitoes can mechanically carry LSDV for up to 8 days. In Chihota's study, during a 2-6 day period after feeding upon lesions of cattle infected with LSDV, mosquitoes were found capable of transmitting the virus. Our study detected LSDV DNA in Culex tritaeniorhynchus Giles, suggesting that Culex tritaeniorhynchus Giles may play an important role in the transmission of LSDV as a vector of mechanical transmission. The aim of our study is to draw attention to the importance of mosquitoes as vectors in the transmission of LSDV. Moreover, we have revised the sentence according to your concern.

The following is a reference to the Sanz-Bernardo and Chihota study.

Sanz-Bernardo B, Suckoo R, Haga IR, Wijesiriwardana N, Harvey A, Basu S, Larner W, Rooney S, Sy V, Langlands Z, Denison E, Sanders C, Atkinson J, Batten C, Alphey L, Darpel KE, Gubbins S, Beard PM. The Acquisition and Retention of Lumpy Skin Disease Virus by Blood-Feeding Insects Is Influenced by the Source of Virus, the Insect Body Part, and the Time since Feeding. J Virol. 2022 Aug 10;96(15):e0075122. doi: 10.1128/jvi.00751-22

Chihota CM, Rennie LF, Kitching RP, Mellor PS. Mechanical transmission of lumpy skin disease virus by Aedes aegypti (Diptera: Culicidae). Epidemiol Infect. 2001 Apr;126(2):317-21. doi: 10.1017/s0950268801005179. PMID: 11349983; PMCID: PMC2869697.

Finally, we appreciate your contribution to contriving our manuscript more appropriately.

This manuscript is a resubmission of an earlier submission. The following is a list of the peer review reports and author responses from that submission.

Round 1

Reviewer 1 Report

Manuscript ID viruses-2098021 -Revision

Manuscript ID viruses-2098021 -Revision

The authors of the manuscript entitled “Detection of Culex Tritaeniorhynchus Giles and novel recombinant strain of lumpy skin disease virus causing high mortality in yaks” report a lumpy skin disease (LSD) outbreak in Sichuan province, China in a mixed farm of yaks and cattle. The authors have very nice data on LDV infection in yaks and presence of virus in faecal samples which is novel and should be published. Unfortunately, at its current stage, the manuscript is not ready for publication. A lot of further work is required to improve the quality and scientific content of the manuscript. Please find below the reasoning for rejection as well as suggestions and encouragement for authors to continue working with the manuscript to improve its quality.

1.    English language needs major improvement, starting from the first sentence of the abstract.

2.    There is a need to correct several factual errors:

·         Be accurate, ticks are not insects, like flies and mosquitos (line 32)

·         African buffaloes are not known to be particularly susceptible or to show clinical signs of LSD (line 34), maybe the authors mean domestic Asian water buffalo?

·         Not only Greece, Serbia and Croatia (line 40) but all the Balkan countries were using homologous vaccine.

·         Armenia vaccinated with sheeppox vaccine (Line 40) (Markosyan et al., J Vet Sci Technol 2017, 8:6, DOI: 10.4172/2157-7579.1000485).

·         Russia vaccinated with sheeppox vaccine, not goatpox (several references available) (Line 41).

·         No details of Lumpivax given Line 47.

·         I would recommend the authors the article about the origin of the recombinant strain: Vandenbussche, F, Mathijs, E. Philips, W, Saduakassova, M., De Leeuw, I., Sultanov, A., Haegeman, A., De Clercq, K. Recombinant LSDV Strains in Asia: Vaccine Spillover or Natural Emergence? Viruses 2022, 14, 1429. https://doi.org/10.3390/v14071429

·         India, Bangladesh and Myanmar did not report the so-called novel recombinant strain (Line 50-51).

3.    Materials and methods -part is poorly written and incomplete, needing re-writing throughout.

4.    Detailed epidemiological description of the outbreak needs to be added to the Material and Methods -part. It should be clearly indicated if the data was collected only from one farm or several farms as it helps to evaluate the morbidity and mortality figures.  

5.     “Sitfast” (Line 96) refers to long lasting skin lesions and not  open ulcers

6.    Culex Tritaeniorhynchus Giles must be always written in italics.

7.    Finding PCR positive Culex Tritaeniorhynchus Giles insects does not provide an absolute proof that these species can transmit the disease, only that insects have been feeding on infected cattle or yaks and may serve as vectors. Experimental transmission studies are required to demonstrate the actual transmission of the of virus from infected to naïve animals by these insects.

8.    Based on the results of one farm and 15 yaks, I’m not convinced that the novel recombinant strain is more virulent in yaks Line 61-62) than the other circulating field strains.

Reviewer 2 Report

Li et al, presents a study where clinical samples were isolated from the yak and cattle in Sichuan province in China were tested for the presence of LSDV. The arthropod Culex spp. was isolated and tested for viral DNA. The genome of the isolated LSDV was sequenced and identified as a novel 'vaccine-like' recombinant strain.

While the study is very relevant to understanding LSDV transmission and pathology in other animals like the yak, the article has serious limitations. 

The abstract mentions high mortality in yaks and cattle but nowhere in the article is epidemiological data presented. I would expect a defined sample number (n=?). Of the denoted number, mortality and morbidity must be described. The methods lacked this crucial information and therefore this conclusion can not be supported. In fact, because this conclusion is not justified by the research undertaken, the title of the article will need to be changed.

I am not going to go through individual spelling and grammatical mistakes but do advise extensive review of English language in the article. Formatting also needs to be looked at. Giles in describing the Culex spp. should not italicised whereas the genus and specific name is specified. 

Reference 9 does not support the statement made and I would advise going through all references. There needs to be appropriate presentation of the facts. Literature says Russia only used sheeppox virus. 

Reference to the 2021 outbreak of LSDV in yak and cattle is missing. If the current manuscript is referring to anecdotal evidence, a reference must still be supplied. The current manuscript does not fully describe this outbreak as noted above.

Methods - again cattle number, farm numbers are important.

Field trappings is not clearly described. Another limitation here is whether you got what was predominant in the area and whether you could detect LSDV DNA in other insect species is unknown.

PCR used and primers not described.

Under whole genome sequencing, was this De Novo assembly or a reference was used? If so, which reference?

Your sample numbers are very few especially insects considering you were trapping field ones.

Fig 4 colours need improvement. The bold colours make the dendrograms difficult to see. Please use accepted and widely used nomenclature for goatpox. In the legend include the software used.

Second before last paragraph was supposed to be mentioned or referred to in the introduction,

The last paragraph is not supported by this study as no data was provided on mortality. 

Reviewer 3 Report

Introduction

*Line 41: Replace used by use

*Line 42-46:  Sentence structure not OK. Please rephrase.

*Line 48: Replace tinue by continue

*Line 49:  “incorrect use of "similarly" --> This would mean that China also used this vaccine or imported animals from Kazakhstan; please rephrase this sentence

*Line 51: No recombinants in India, Bangladesh and Myanmar -> NI2490 like

*Line 61: “morbidity”: This is confusing because the sentence above is about the mortality and here now of morbidity

Material and methods

*Line 67: “typical clinical signs”: can you please indicate which typical clinical signs

*Line 69-70: are the samples from the same animals? This section is confusing. The results section states that there is 100% morbidity with 15 / 15 yaks. Why this limited number of samples? And why only 3 skin nodules? Did the other animals have no nodules then? Why marked as LSDV anyway?

*Line 74: Please start title with capital letter “Whole genome sequencing”

Sentence structure not OK. Please rephrase.

Results and discussions

*Line 91-92: Samples and Province: no capital letter

*Line 99: Positivity rate for fecal swabs and skin nodules is 75-100% and not 31.25-100%

*Line 100: Can you please indicate what you mean with high levels?

*Line 102: what you mean with “in addition”? This study?

*Line 102-104: If the results are only based on this study (1 farm with 15 yaks), they need to be nuanced more as generalization on these limited data is risky

*Line 112- 114: Sentence structure not ok

*Line 134-136: this needs to be nuanced, indirect evidence that this vector plays an important role in the transmission of LSDV.